# Simple Denoising Diffusion Language Models

**Huaisheng Zhu** [1]   **Zhengyu Chen** [2]   **Shijie Zhou** [3]   **Zhihui Xie** [4]   **Yige Yuan** [5]   **Shiqi Chen** [6]   **Zhimeng Guo** [1]
**Siyuan Xu** [1]   **Hangfan Zhang** [1]   **Vasant Honavar** [3]   **Teng Xiao** [7][8]

## Abstract

Recent Uniform-state Diffusion Models (US-DMs), initialized from a uniform prior, offer the promise of fast text generation due to their inherent self-correction ability compared to masked diffusion models. However, they still rely on complex loss formulations with additional computational overhead, which hinders scalability. In this work, we explore a simplified denoising-based loss for USDMs that optimizes only noise-replaced tokens, stabilizing training while matching the performance of prior methods with more complex objectives. In addition, we introduce an efficient regularization term to mitigate corruption toward uniform output distributions, which further improves performance. We demonstrate the effectiveness and efficiency of our simple and improved loss formulations by pretraining models on widely used text datasets for USDMs. More importantly, our conclusions scale to larger models, showing strong potential for large-scale training. The code of our method is available at this link.

## 1. Introduction

Diffusion models are powerful generative frameworks that excel at producing realistic, high-quality continuous data such as images and videos (Ho et al., 2020; Song et al., 2020a; Rombach et al., 2022; Kong et al., 2020; Ho et al., 2022). They achieve this by training denoising models to reconstruct samples corrupted with varying levels of Gaussian noise. Generation then proceeds through a Markov chain: starting from pure noise, the model iteratively denoises the sample, gradually transforming it into a clean image.

To further advance the capabilities of diffusion models,

---

[1]Penn State University [2]Meituan [3]University at Buffalo [4]The University of Hong Kong [5]Alibaba Group [6]City University of Hong Kong [7]University of Washington [8]Allen Institute for AI (AI2). Correspondence to: Huaisheng Zhu <hvz5312@psu.edu>.

*Proceedings of the 43rd International Conference on Machine Learning*, Seoul, South Korea. PMLR 306, 2026. Copyright 2026 by the author(s).

Masked Diffusion Models (MDMs), which use the prior distribution by masking all tokens, have recently demonstrated remarkable progress across language generation tasks (Sohl-Dickstein et al., 2015; Austin et al., 2021; Campbell et al., 2022; Lou et al., 2023; Meng et al., 2022). By optimizing the simplified varaint of evidence lower bound (ELBO), masked diffusion language models have achieved performance comparable to, and in some cases surpassing, that of autoregressive models (ARMs) (Sahoo et al., 2024; Shi et al., 2024; Nie et al., 2025). Moreover, recent studies have investigated the scaling properties of MDMs, demonstrating that they can achieve competitive performance with advanced autoregressive models of similar size (e.g., Llama 2 (Touvron et al., 2023) and Llama 3 (Dubey et al., 2024)) on a range of downstream tasks (Nie et al., 2025; Gong et al., 2024; Nie et al., 2024; Gong et al., 2025; Ye et al., 2025).

Despite its great success, MDMs experience severe performance degradation in the few-step regime (Deschenaux & Gulcehre, 2024). In contrast to diffusion models with Probability Flow ODEs in continuous space (Song et al., 2020b), MDMs lack an implicit property—a deterministic mapping from noise to data. To address this limitation, recent work on Uniform State Diffusion Models (USDMs) explore language modeling by initializing from a uniform distribution, analogous to Gaussian noise in continuous diffusion models (Sahoo et al., 2025; Austin et al., 2021; Zhao et al., 2024; Schiff et al., 2024). Inspired by the extensive success of Gaussian diffusion, these models benefit from a range of well-studied techniques, especially efficient training and distillation schemes that enable fast generation (Song et al., 2023). These models achieve performance comparable to MDMs while demonstrating strong potential to reduce the number of sampling steps without compromising generation quality. Moreover, USDMs exhibit superior scaling behavior compared to MDMs as training FLOPs increase, demonstrating their promise as a direction for diffusion-based language models (von Rütte et al., 2025).

However, the current state-of-the-art USDMs (Sahoo et al., 2025), which adopts uniform distributions as the prior, still suffers from a complex loss formulation. This complexity leads to additional computational overhead during training and may hinder scalability. Therefore, in this paper, it naturally inspires us to *explore how to simplify the training*

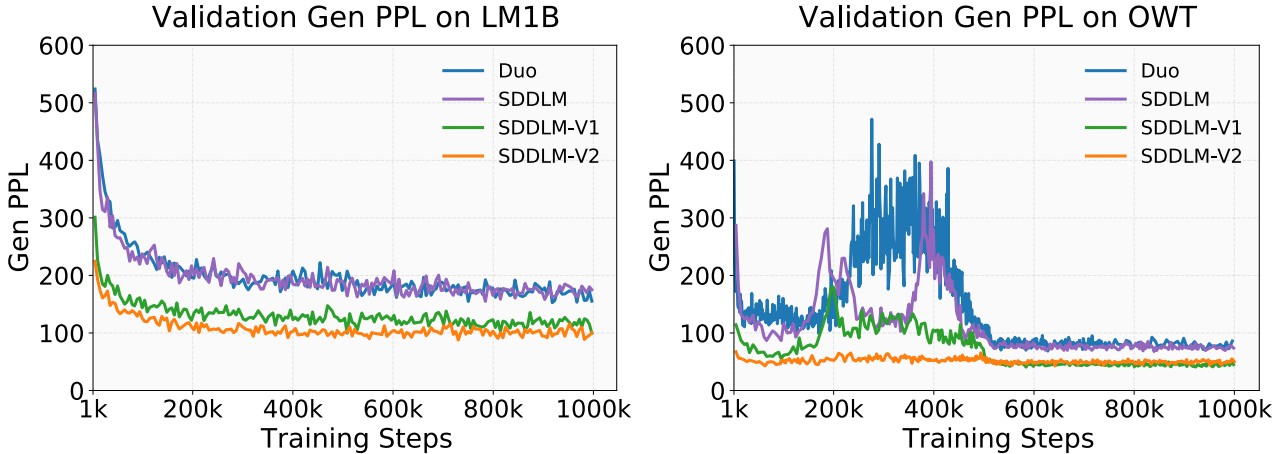

*Figure 1.* Validation Gen PPL of different models over training steps.

*objective for USDMs with great scaling abilities.*

To resolve this problem, we explore a simpler loss formulation for diffusion language models initialized from a uniform prior (i.e., pure noise). Specifically, analogous to diffusion models with Gaussian noise in continuous spaces (Ho et al., 2020), we start from standard denoising objectives, in which noise is added to clean sequences and the model is trained to denoise the corrupted inputs back to the original sequences. However, we find that this naive adaptation frequently leads to training collapse. Motivated by the design of MDMs, we instead propose a method called Simple Denoising Diffusion Language Model (SDDLM), which optimizes only the tokens that are replaced with noise. This strategy, akin to the selective denoising behavior of MDMs, stabilizes training and achieves performance comparable to the ELBO-derived loss, while avoiding its significant computational cost. Moreover, we introduce Anti-Uniform Distribution Sharpening regularization to mitigate prediction corruption caused by noisy sequences that tend to induce near-uniform output distributions. From a self-supervised learning perspective, the resulting negative gradient is closely related to contrastive learning (Xiang et al., 2023; Chen et al., 2024). As shown in Figure 5, our results with SDDLM-V1 demonstrate that this design yields notable improvements in generation quality.

**Contributions.** The main contributions of this paper are as follows: (i) We introduce a *simple and efficient denoising-based training framework* for USDMs that substantially simplifies prior training objectives while retaining strong empirical performance. (ii) We further improve performance by proposing a *regularization term with negative gradients* (SDDLM-V1), motivated by a self-supervised and contrastive learning interpretation of the denoising objective, which mitigates degeneration toward uniform output distributions. (iii) We demonstrate that SDDLM achieves *competitive performance* relative to prior ELBO-based ap-

proaches that incur higher computational overhead during training. Also, SDDLM-V1 consistently improves generation quality. Importantly, these results *scale to larger models (up to 1.1B parameters)*, indicating indicating strong effectiveness for large-scale pretraining.

## 2. Background

### 2.1. Denoising Diffusion Models

Denoising diffusion models on continuous space formulate generation as a Markov process that transforms the data distribution $q_{\text{data}}$ into a simple prior on continuous space, such as a standard normal distribution $\mathcal{N}(0, \mathbf{I})$. Concretely, the process begins with samples from the data distribution and iteratively adds noise to produce a sequence of noisy latents $\mathbf{x}_t \sim q_t(\cdot \mid \mathbf{x}_0)$, whose marginal distribution is:

$$q_t\left(\cdot \mid \mathbf{x}_0; \bar{\alpha}_t\right) = \mathcal{N}\left(\mathbf{x}_t; \sqrt{\bar{\alpha}_t}\mathbf{x}_0, (1 - \bar{\alpha}_t)\,\mathbf{I}\right), \quad (1)$$

where the diffusion parameter $\bar{\alpha}_t \in [0, 1]$ is a monotonically decreasing function in $t$ and $\mathbf{x}_0 \sim q_{\text{data}}$. Then, a simplified version of evidence lower bound (ELBO) based on denosing noisiy images into clean images is minimized to train the diffusion model with the following equation:

$$\mathbb{E}_{\mathbf{x}_0, t, \epsilon}\left[\lambda(t)\left\|\mathbf{x}_0 - \mathbf{x}_\theta\left(\mathbf{x}_t, t\right)\right\|^2\right] \quad (2)$$

where $\epsilon \sim \mathcal{N}(0, \mathbf{I}), t \sim \mathcal{U}(0, T), \mathbf{x}_t \sim q_t\left(\cdot \mid \mathbf{x}, \bar{\alpha}_t\right)$. $\lambda(t)$ is a time dependent weighting function and can be ignored during the training process. $\theta$ are learnable parameters.

### 2.2. Discrete Diffusion Models

Previous objectives and formulations are primarily based on Gaussian distributions and operate in continuous spaces. To adapt diffusion models to discrete data $\mathbf{x} \in \mathcal{V}$, where $\mathcal{V}$ denotes the vocabulary for language generation, the discrete diffusion framework (Sohl-Dickstein et al., 2015; Austin

et al., 2021) extends the core idea of continuous denoising diffusion models: mapping the data distribution $q_{\text{data}}$ to a simple prior distribution through a sequence of Markov states. Similar to their continuous counterparts, the noise-adding process—referred to as the forward process $(q_t)(t \in [0, 1])$—smoothly transitions from $q_{\text{data}}$ to a categorical prior $\text{Cat}(\cdot; \boldsymbol{\pi})$ by interpolating between the data distribution and the prior. The corresponding marginals conditioned on one token $\mathbf{x}_0^l$ at time $t$ are given by the following equation:

$$q_t\left(. \mid \mathbf{x}_0^l; \alpha_t\right) = \text{Cat}\left(.; \alpha_t \mathbf{x}_0^l + (1 - \alpha_t)\,\boldsymbol{\pi}\right). \quad (3)$$

For MDLMs, the prior $\boldsymbol{\pi}$ is typically defined using a special masked token, i.e., $\boldsymbol{\pi} = \mathbf{M}$ with $\mathbf{M} \in \mathcal{V}$ (Sahoo et al., 2024). Alternatively, a uniform prior can be defined as $\boldsymbol{\pi} = \mathbf{1}/V$, where $V = |\mathcal{V}|$ denotes the vocabulary size. Specifically, in MDMs, a token $\mathbf{x}$ either stays unchanged or is replaced by the mask token $\mathbf{m}$, remaining masked thereafter. In USDMs, each token can instead transition uniformly to any token in $\mathcal{V}$, with probabilities determined by the diffusion timestep. To train USDMs, the Negative Evidence Lower Bound (NELBO) loss for the token $l$ is derived using principles similar to those of continuous diffusion models, and can be expressed in the following form (Lou et al., 2023; Schiff et al., 2024; Sahoo et al., 2025):

$$\mathcal{L}_{\text{USDM}}^l = \mathbb{E}_{t \sim \mathcal{U}[0,1], q_t\left(\mathbf{x}_t^l \mid \mathbf{x}_0^l; \alpha_t\right)} - \frac{\alpha_t'}{V \alpha_t} \left[ \frac{V}{\tilde{\mathbf{x}}_i^l} \right.$$
$$\left. - \frac{V}{\left(\tilde{\mathbf{x}}_\theta^l\right)_i} - \sum_j \frac{\tilde{\mathbf{x}}_j^l}{\tilde{\mathbf{x}}_i^l} \log \frac{\left(\tilde{\mathbf{x}}_\theta^l\right)_i \cdot \tilde{\mathbf{x}}_j^l}{\left(\tilde{\mathbf{x}}_\theta^l\right)_j \cdot \tilde{\mathbf{x}}_i^l} \right], \quad (4)$$

where $\tilde{\mathbf{x}}^l = V \alpha_t \mathbf{x}_0^l + (1 - \alpha_t)\,\mathbf{1}$, $\tilde{\mathbf{x}}_\theta^l = V \alpha_t \mathbf{x}_\theta^l(\mathbf{x}_t, t) + (1 - \alpha_t)\,\mathbf{1}$, $\mathbf{x}_t^l \sim q_t(\cdot \mid \mathbf{x}_0^l; \alpha_t)$ and $\alpha_t'$ is the time-derivative of the $\alpha_t$. $i = \arg\max_{j \in [V]}\left(\mathbf{x}_t^l\right)_j$ is the non-zero entry of $\mathbf{x}_t^l$. Other studies have also explored more efficient approaches to computing the loss in Equation (4) (Sahoo et al., 2025). $\mathbf{x}_\theta^l$ denotes a neural network $\mathcal{V} \times [0, 1] \to \Delta^V$, where $\Delta^V$ denotes the K-simplex and $\theta$ are trainable parameters . After training models with Equation (4), USDMs typically generate samples by applying the reverse diffusion process, starting from the uniform prior in the following equation:

$$q_{s|t}\left(. \mid \mathbf{x}_t^l, \mathbf{x}_0^l\right) = \text{Cat}\left( .; \frac{V \alpha_t \mathbf{x}_t^l \odot \mathbf{x}_0^l + \left(\alpha_{t|s} - \alpha_t\right)\mathbf{x}_t^l}{V \alpha_t \left\langle \mathbf{x}_t^l, \mathbf{x}^l \right\rangle + 1 - \alpha_t} \right.$$
$$\left. + \frac{\left(\alpha_s - \alpha_t\right)\mathbf{x}_0^l + \left(1 - \alpha_{t|s}\right)\left(1 - \alpha_s\right)\mathbf{1}/V}{V \alpha_t \left\langle \mathbf{x}_t^l, \mathbf{x}_0^l \right\rangle + 1 - \alpha_t} \right), \quad (5)$$

where $s < t$, $\alpha_{t|s} = \alpha_t/\alpha_s$ and $\mathbf{x}^l$ represents a one-hot vector, with a nonzero entry indicating the token. During inference, we replace $\mathbf{x}$ in Equation (5) with $\mathbf{x}_\theta(\mathbf{x}_t, t)$. In the following section, we use $p_\theta(\mathbf{x}_0 \mid \mathbf{x}_t)$ to denote $\mathbf{x}_\theta(\mathbf{x}_t, t)$.

## 3. Related Works

**Denoising Diffusion Models.** Denoising diffusion probabilistic models have proven to be powerful tools for generating diverse data types (Ho et al., 2020; Song et al., 2020a). The sampling process of diffusion models can be interpreted as stochastic differential equations (SDEs) and is trained using score matching objectives based on this formulation (Song et al., 2020b). Moreover, the strong generative performance of Denoising Diffusion Models (DDMs) has attracted increasing interest in their potential for representation learning (Xiang et al., 2023; Chen et al., 2024), as their training process is equivalent to that of Denoising Autoencoders (DAEs) (Vincent et al., 2008), which remove noise at multiple levels through a diffusion-driven procedure. Interestingly, aligning denoising diffusion models with self-supervised learning–based models enables efficient training of generative diffusion models (Yu et al., 2024). This insight motivates us to reconsider the training objective for USDMs from a self-supervised learning perspective, where a denoising objective enables diffusion models to be trained in a continuous space. Moreover, contrastive learning (Chen et al., 2020), by providing informative negative gradients, can further enhance representation learning, which in turn inspires us to further explore training objectives for USDMs.

**Diffusion Lanuage Models.** The development of DLLMs is motivated by recent advances in discrete diffusion models, which introduced new forward and reverse transition mechanisms and enabled a diverse range of model variants (Sohl-Dickstein et al., 2015; Austin et al., 2021; Campbell et al., 2022; Lou et al., 2023; Meng et al., 2022). Empirical studies further demonstrate that masked diffusion models (MDMs) can achieve perplexity comparable to autoregressive models (ARMs) (Sahoo et al., 2024; Shi et al., 2024; Nie et al., 2025; Ou et al., 2024). To improve training efficiency, several works have proposed simplified training objectives for masked diffusion processes with theoretical justifications. In addition, recent research has examined the scaling behavior of MDMs, including both training from scratch and adaptation from pre-trained ARMs (Nie et al., 2025; Gong et al., 2024; Nie et al., 2024; Ni et al., 2025b;a). Although MDLMs demonstrate greater efficiency than ARMs by generating multiple tokens simultaneously, they suffer from notable performance degradation in the few-step generation regime (Deschenaux & Gulcehre, 2024). While numerous techniques for reducing sampling steps without sacrificing generation quality have been successful in continuous-space diffusion models, directly transferring these methods to MDMs is difficult, as they lack the inherent property of mapping noise to data. To overcome this limitation, recent works on Uniform-state Diffusion Models (USDMs) have explored initializing language models from a uniform distribution, analogous to Gaussian noise in continuous diffusion (Sahoo et al., 2025; Austin et al., 2021; Zhao et al.,

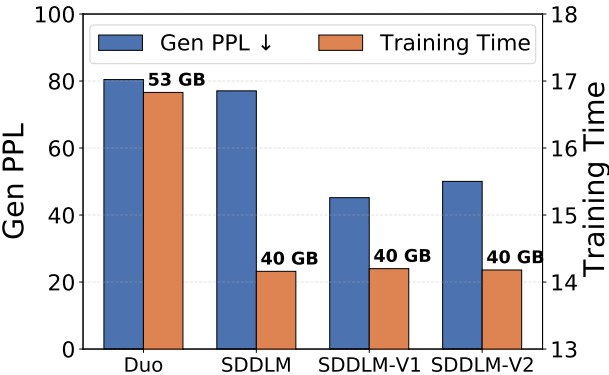

*Figure 2.* Running time, memory and generation quality comparison on OWT. We report the training time for 0.5 epochs and the peak GPU memory for each method under the same batch size, together with their generation quality.

2024; Schiff et al., 2024). Unlike MDLMs, which simplify the training loss and scale effectively to larger models, current USDMs still rely on complex ELBO-derived losses, potentially limiting their scalability. Therefore, in this paper, we study the problem of simplifying the loss of USDMs.

# 4. Method

In this section, we investigate the use of a simple denoising loss to improve USDMs. We first observe that a naive application of this loss can lead to performance degradation. To address this issue, we apply the denoising loss only to perturbed positions, which effectively mitigates the observed degradation, denoted as Simple Denoising Diffusion Language Model (SDDLM). Building on this, we further introduce an anti-uniform distribution sharpening regularization that incorporates negative gradients and denote this method as SDDLM-V1. This regularization counteracts the tendency toward uniform distribution corruption during training, leading to more stable optimization and improved final performance. Finally, we connect our design with contrastive learning and inspire other design in the future work.

## 4.1. Simple Denoising Diffusion Language Models

Comparing the training loss of continuous diffusion models in Equation (2) with that of USDMs in Equation (4), we observe that the latter is considerably more complex. To address this, we propose a simplified formulation of the USDMs loss objective. First, we consider optimizing an objective analogous to the reconstruction loss in Equation (2), where the model directly learns to predict the original sequence (the desired target) by replacing the clean sequence $\mathbf{x}_0$ in Equation (5). This approach is also used in training

discrete flow-matching models (Gat et al., 2024):

$$\min_\theta \mathbb{E}_{\mathbf{x}_0,t,\mathbf{x}_t \sim \mathcal{D},\mathcal{U}[0,1],q_t} \sum_{l=1}^{L} -\log p_\theta(\mathbf{x}_0^l \mid \mathbf{x}_t), \quad (6)$$

where $\mathbf{x}_0$ are sequences of tokens from $\mathcal{D}$ and $\mathbf{x}_t$ is used to reconstruct $\mathbf{x}_0$ with a denoising loss and each token of $\mathbf{x}$ is perturbed by Equation (3). However, training with this loss leads to degraded model performance and unstable optimization. Note that we exclude this baseline from comparisons due to unstable convergence. We attribute this issue to the structure of the sequence $\mathbf{x}_t$, which consists of two parts: (i) positions where $\mathbf{x}_t^j \neq \mathbf{x}_0$, corresponding to tokens corrupted by noise, and (ii) positions where $\mathbf{x}_t^j = \mathbf{x}_0$, which remain unchanged. When applying a reconstruction-based objective, these two parts impose different learning goals: the noisy positions require denoising, while the unchanged positions reduce to reconstructing the input itself. During our empirical studies, we find that the denoising component is more critical, as the model must reconstruct clean sequences through its denoising ability. This process is also closely aligned with the training paradigm of MDMs, which predict only the masked tokens; this perspective also inspires us to focus prediction on the noisy components. Guided by this observation, we propose focusing the objective on the denoising part of the sequence by the following equation:

$$\mathcal{L}_{\text{SDDLM}} =$$
$$\mathbb{E}_{\mathbf{x}_0,t,\mathbf{x}_t \sim \mathcal{D},\mathcal{U},q_t} \sum_{l=1}^{L} -\log p_\theta(\mathbf{x}_0^l \mid \mathbf{x}_t)\mathbf{1}\left[\mathbf{x}_0^l \neq \mathbf{x}_t^l\right]. \quad (7)$$

This simple objective achieves generation performance comparable to the original, more complex NEBLO loss in Equation (4), as shown in Figure 1. Moreover, it incurs substantially lower training cost while maintaining comparable performance, as illustrated in Figure 2. The training details of algorithms are put into Appendix C.4. Moreover, to further clarify the connection between our simple algorithms and the ELBO objective used in prior methods, we provide a detailed analysis of the relationship between our loss and the ELBO loss in Section A.2 following Zekri et al. (2026).

## 4.2. Anti-uniform Distribution Sharpening

So far, we have focused on a simple denoising objective to improve training efficiency. We now explore whether this formulation can be further improved in terms of overall performance, while retaining its simplicity. USDMs employ a uniform prior over the vocabulary, which introduces strong stochasticity during the corruption process compared with AR models and MDMs. This simple denoising loss induces a characteristic failure mode: under heavy corruption, the conditional distribution tends to become overly smooth and close to uniform. In such cases, a standard denoising objective that only rewards the ground-truth token

| | BoolQ | Hellaswag | Obqa | PIQA | RACE | SIQA | LAMBADA |
|---|---|---|---|---|---|---|---|
| SDDLM | 60.34 | 34.60 | 18.20 | 64.42 | 30.05 | 37.56 | 26.53 |
| SDDLM-CFG | **62.11** | **36.09** | **20.00** | **64.80** | **30.72** | **37.87** | **32.60** |

*Table 1.* Results for CFG methods on 1.1B models.

may provide insufficient discriminative signal, leading to high-entropy predictions and degraded generation quality. To mitigate this effect, we introduce an anti-uniform distribution sharpening regularizer that explicitly encourages the model's conditional prediction to deviate from the uniform distribution. Concretely, in addition to maximizing the likelihood of the ground-truth token, we penalize probability mass assigned to randomly sampled tokens from the vocabulary. The resulting objective is shown as follows:

$$\mathcal{L}_{\text{SDDLM}} - \sum_{l=1}^{L} \text{KL}\left(\mathcal{U} \| p_\theta\left(\mathbf{x}_0^l \mid \mathbf{x}_t\right)\right) \mathbf{1}\left[\mathbf{x}_0^l \neq \mathbf{x}_t^l\right], \quad (8)$$

where $\mathcal{U}$ denotes the uniform distribution over the vocabulary $\mathcal{V}$, i.e., $\mathcal{U} = 1/|\mathcal{V}|$ for all $v \in \mathcal{V}$. Thus, minimizing this objective implicitly maximizes the discrepancy between the uniform distribution and the model's predictions, encouraging the model to move away from uniform outputs and produce sharper, more discriminative token distributions. This effect counteracts the over-smoothing induced by uniform-state corruption. After simple mathematical derivation (in Appendix A.1), we get the following equation:

$$\mathcal{L}_{\text{SDDLM-V1}} =$$
$$\mathbb{E}_{\mathbf{x}_0, t, \mathbf{x}_t \sim \mathcal{D}, \mathcal{U}, q_t} \sum_{l=1}^{L} (-\log p_\theta(\mathbf{x}_0^l \mid \mathbf{x}_t) \quad (9)$$
$$+ \mathbb{E}_{\hat{\mathbf{x}}^l \sim \text{U}(\mathcal{V})} \log p_\theta(\hat{\mathbf{x}}^l \mid \mathbf{x}_t)) \mathbf{1}\left[\mathbf{x}_0^l \neq \mathbf{x}_t^l\right],$$

where $\hat{\mathbf{x}}^l \sim \text{U}(\mathcal{V})$ denotes a token randomly sampled from the vocabulary. However, during optimization, the negative gradient may dominate due to the large gradients induced by small values in the logarithmic terms, leading to training instability and model degradation. We add a small constant $\varepsilon$ to $p_\theta(\mathbf{x}_0^l \mid \mathbf{x}_t)$ and $p_\theta(\hat{\mathbf{x}}_0^l \mid \mathbf{x}_t)$ to stabilize the gradient of the logarithm. Note that the training of SDDLM without adding $\epsilon$ will directly lead to break for the model so we ignore this training ablation studies in our experiment section. Our proposed loss objective is a simple modification of Equation (7) and proves to be practical and effective.

### 4.3. Connection with Contrastive Learning

We propose a simple regularization term that substantially improves generation quality, as demonstrated by our empirical results. In this section, we further analyze and seek to understand the effects of this regularization term from other perspectives. Our first SDDLM adopts a denoising objective from a self-supervised learning perspective. We further

interpret our improved version, SDDLM-V1 through the lens of contrastive learning (Chen et al., 2020), a prominent self-supervised paradigm, by analyzing its gradient structure. This structure comprises both attractive and repulsive gradient components, which are known to be effective for representation learning. From this perspective, SDDLM-V1 naturally incorporates both attractive and repulsive gradients, a property that may also be beneficial for leveraging negative gradients. In particular, the gradient of our loss (Equation (9)) takes the following form, which closely resembles that used in contrastive learning:

$$-\nabla_\theta z_\theta\left(\mathbf{x}_0^l\right) + \mathbb{E}_{\hat{\mathbf{x}}^l \sim \text{U}(\mathcal{V})} \nabla_\theta z_\theta\left(\hat{\mathbf{x}}^l\right), \quad (10)$$

where $z_\theta = \log p_\theta(\mathbf{x}_0^l \mid \mathbf{x}_t)$. This perspective from contrastive learning provides a useful interpretation of our method and suggests potential avenues for further improvement through self-supervised learning perspectives in future. Therefore, this insight further motivates SDDLM-V1 to design more principled negative sampling strategies beyond random sampling from a uniform distribution.

### 4.4. Classifier-free Guidance on SDDLM

Classifier-Free Guidance (CFG) (Ho et al., 2020) is an effective and versatile technique widely used in both continuous and discrete diffusion models (Ho & Salimans, 2022; Lovelace et al., 2024). Grounded in Bayes' rule, CFG jointly trains conditional and unconditional diffusion models and introduces a rescaled distribution during inference to guide generation. However, prior methods require the conditional model to be trained on paired data (prompt–response pairs) before applying classifier-free guidance (CFG). In practice, during language model pretraining, it is difficult to obtain sufficient amounts of such conditional data. To address this limitation, we adopt unsupervised CFG (Nie et al., 2024) and use randomly sampled tokens as the conditioning input for the unconditional generation branch:

$$\tilde{p}_{\boldsymbol{\theta}}\left(\boldsymbol{x}_0 \mid \boldsymbol{c}, \boldsymbol{x}_t\right) \propto \frac{p_{\boldsymbol{\theta}}\left(\boldsymbol{x}_0 \mid \boldsymbol{c}, \boldsymbol{x}_t\right)^{1+w}}{p_{\boldsymbol{\theta}}\left(\boldsymbol{x}_0 \mid \boldsymbol{r}, \boldsymbol{x}_t\right)^w}, \quad (11)$$

where $\mathbf{c}$ denotes the prompt for conditional text generation, and $\mathbf{r}$ is a sequence of tokens randomly sampled from the uniform distribution over the vocabulary $\mathcal{V}$. $\tilde{p}_\theta(\cdot)$ denotes the modified distribution that replaces the original $p_\theta$ for likelihood computation and generation in Equations (4) and (5). The results of the CFG methods are reported in Table 1, where we observe that CFG effectively improves the performance of SDDLM on different datasets.

# 5. Experiment

## 5.1. Experimental Setup

**Datasets and Models.** We evaluate our proposed method, SDDLM, along with its negative-gradient variants, SDDLM-V1, on standard language modeling benchmarks: LM1B (Chelba et al., 2013) and OpenWebText (OWT) (Gokaslan et al., 2019). All small models (170M-parameter) are trained for 1M steps for LM1B and OpenWebText with a batch size of 512. For LM1B, we adopt a context length of 128, while for OWT we use a context length of 1024. Moreover, to further validate the effectiveness of our method, we scale up our models to a larger parameter regime (1.1B parameters) and evaluate them on larger-scale datasets. Specifically, we employ the open-source SlimPajama dataset (Shen et al., 2023), a multi-corpus collection containing 627 billion tokens, which is sufficiently large for all of our experiments. For fairness, we use the Llama-2 tokenizer (Touvron et al., 2023) across all models and baselines, and set the context length to 2048.

**Implementation Details.** For small models, we follow the implementation of Duo (Sahoo et al., 2025), including the time-scheduling strategy proposed in their model. Similar to Duo, our architecture is a 170M-parameter modified Diffusion Transformer (DiT (Peebles & Xie, 2023)) with rotary positional encodings (Su et al., 2024) and adaptive layer normalization for conditioning on diffusion time, consistent with prior work. Training is performed on 8×H800 GPUs using bfloat16 precision. For large models, following the implementation of Nie et al. (2024), we adopt Pre-LayerNorm with RMSNorm (Zhang & Sennrich, 2019) for better stability, use SwiGLU (Shazeer, 2020) as the activation function to enhance non-linearity, and implement RoPE (Su et al., 2024) for more expressive positional encoding. Training of large models is performed on 16×H800 GPUs using bfloat16 precision. In the following section, we use the state-of-the-art uniform state diffusion model Duo (Sahoo et al., 2025) as our baseline. We denote our loss with the negative gradient defined in Equation (9), as SDDLM-V1. Moreover, in addition to random sampling, we also use the noisy version $x_t$ itself as negative samples by denoting it as SDDLM-V2.

**Evaluation.** For small models, our zero-shot datasets include the validation splits of WikiText (Merity et al., 2016), Lambda (Paperno et al., 2016), AG News (Zhang et al., 2015), and Scientific papers from ArXiv and Pubmed (Cohan et al., 2018). For large models, to provide a comprehensive evaluation for large models, we assess SDDLMs on eight widely used benchmarks in the zero-shot setting, covering tasks in commonsense reasoning and reading comprehension: Hellaswag (Zellers et al., 2019), ARC-e (Clark et al., 2018), BoolQ (Clark et al., 2019), PIQA (Bisk et al., 2020), SIQA (Sap et al., 2019), Obqa (Mihaylov et al., 2018), RACE (Lai et al., 2017), and LAMBADA (Paperno

| Model | LM1B | | OWT | |
|---|---|---|---|---|
| | Gen PPL ↓ | Entropy ↑ | Gen PPL ↓ | Entropy ↑ |
| Duo | 172.93 | **4.20** | 80.43 | **5.55** |
| SDDLM | 173.04 | **4.20** | 77.07 | 5.53 |
| SDDLM-V1 | 116.84 | 4.10 | **45.18** | 5.31 |
| SDDLM-V2 | **101.32** | 4.12 | 50.05 | 5.33 |

*Table 2.* Comparison of Gen PPL and Entropy across LM1B and OWT on SDDLM, SDDLM-V1, SDDLM-V2 and Duo.

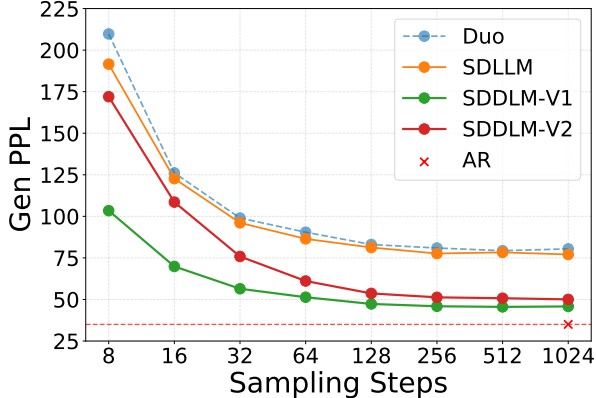

*Figure 3.* Sample quality comparison of SDDLM and its variant vs. Duo. SDDLM-V1 outperforms Duo in Gen PPL (↓) for base models across different sampling steps.

et al., 2016). Details of datasets are put into Appendix C.1.

**Baseline.** The primary baselines for our method are the state-of-the-art Duo (Sahoo et al., 2025), USDMs (SEDD Uniform (Lou et al., 2023), denoted as SEDD for simplicity, and UDLM (Schiff et al., 2024)) and Gaussian diffusion method, PLAID (Gulrajani & Hashimoto, 2023). For a fair comparison, we directly adopt the implementation with the best hyperparameters and reported results from the original paper. The details of all baselines are put into Appendix C.2.

## 5.2. Sample Quality Comparison

To evaluate sample quality, we report GPT-2 Large generative perplexity (Gen PPL) as a measure of fluency and average sequence entropy as an indicator of diversity. The corresponding results are presented in Figure 1 and 5 on LM1B for validation over different training steps. Specifically, we first sample a small subset to validate the Gen PPL and entropy metrics shown in these figures, using 1,000 sampling steps. We then report the final results based on a larger set of samples generated from the final checkpoint with 1,024 sampling steps, as presented in Table 2. We observe that our proposed denoising loss (Equation (7)) achieves performance comparable to the baseline in terms of both Gen PPL and entropy. Furthermore, incorporating the negative gradient loss significantly improves Gen PPL, indicating stronger alignment with real-world generative

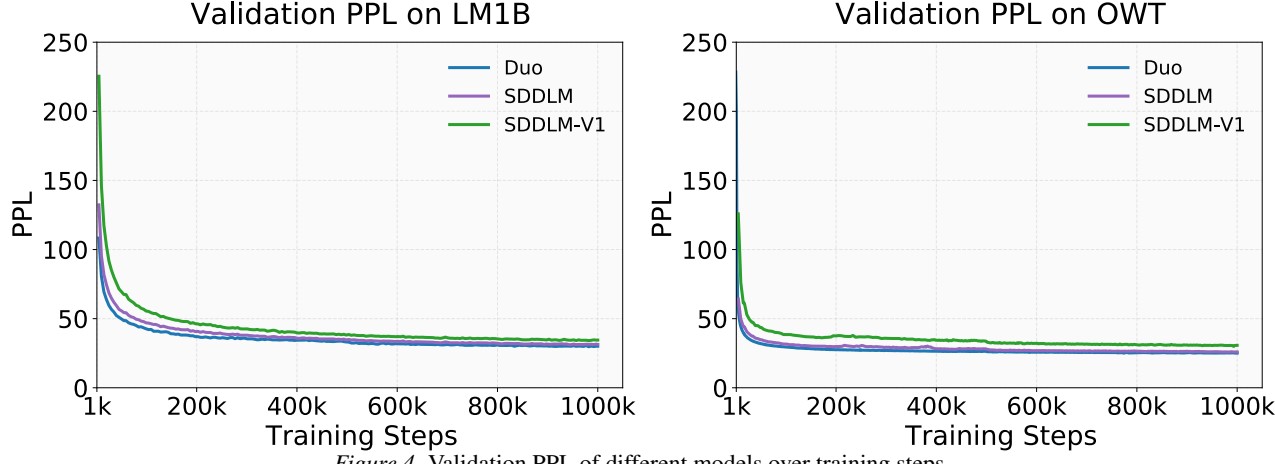

*Figure 4.* Validation PPL of different models over training steps.

quality. These findings suggest that this simple adaptation is effective for generation quality. In Section 5.4, we also further explore the role of negative gradients in enhancing performance on larger models and more complex tasks.

In addition, we evaluate our method under different numbers of sampling steps, as shown in Figure 3. We observe that SDDLM, despite its high training efficiency, achieves performance comparable to approaches based on more complex ELBO-style objectives such as state-of-the-art baselines (Duo) across a range of sampling steps. Furthermore, the proposed regularization term consistently improves generation quality at different sampling steps. These results further demonstrate the strong generalization of our simple training objective and its enhanced variant with additional regularization across different numbers of sampling steps. Moreover, the negative-gradient–based methods show promise for reducing the required number of sampling steps (achieve better performance than Duo while using fewer sampling steps), thereby enabling the generation of more tokens within the same computational budget and improving the overall efficiency of USDMs.

Finally, we report the training cost on the OWT dataset in Figure 2. Our results show that incorporating the negative gradient leads to both improved performance and reduced training cost, measured in terms of wall-clock training time and GPU memory consumption. In addition, despite its simplicity, SDDLM with our proposed loss achieves performance comparable to Duo trained with the ELBO objective. Taken together with the previous observations, these results demonstrate that our method provides a promising and scalable approach for training USDMs.

### 5.3. Likelihood Evaluation

In this section, we estimate the negative log-likelihood using the ELBO, following the formulation in Duo Models (Sa-

|  | Wikitext | Lambada | AG News | Pubmed | Arxiv |
|---|---|---|---|---|---|
| SEDD [†] | 41.10 | 57.29 | 82.64 | 55.89 | 50.86 |
| Plaid[†] | 50.86 | 57.28 | - | - | - |
| UDLM[†] | 39.42 | 53.57 | 80.96 | 50.98 | 44.08 |
| Duo[‡] | **33.91** | **51.29** | **69.71** | 45.34 | 40.41 |
| **SDDLM** | 35.02 | 51.50 | 73.44 | **44.75** | **39.95** |

*Table 3.* Zero-shot perplexities ($\downarrow$) of models trained for 1M steps on OWT datasets. All perplexities for diffusion models are upper bounds. [†] Taken from Sahoo et al. (2025). Best uniform / Gaussian diffusion values are **bolded** and second best values are underlined. [‡] denotes retrained model.

hoo et al., 2025). The corresponding results across different training steps are illustrated in Figure 4. We observe that our proposed denoising loss (Equation (7), SDDLM) and negative gradient with random sampling (SDDLM-V1) lead to a slight increase in perplexity (PPL) when evaluated under the ELBO framework. It is normal that our model does not directly optimize the likelihood, as similar observations have been reported in previous works on masked diffusion language models (Deschenaux & Gulcehre, 2024). Moreover, this discrepancy is not necessarily correlated with generation quality — in fact, we observe a substantial improvement in the quality of generated samples. Interestingly, when applying negative sampling on the perturbed sequence $\mathbf{x}_t$ (SDDLM-V2), the ELBO-based PPL decreases substantially, yet the sampling quality improves significantly. This intriguing phenomenon suggests that optimizing purely for ELBO-based likelihood may not fully capture generation quality. We further investigate this behavior on larger models and more complex tasks in Section 5.4.

Moreover, we measure the zero-shot generalization of the models trained on OWT by evaluating their PPL on 5 other datasets. The corresponding results are shown in Table 3. We observe that SDDLM, which exhibits high training efficiency, achieves exact perplexity (PPL) performance com-

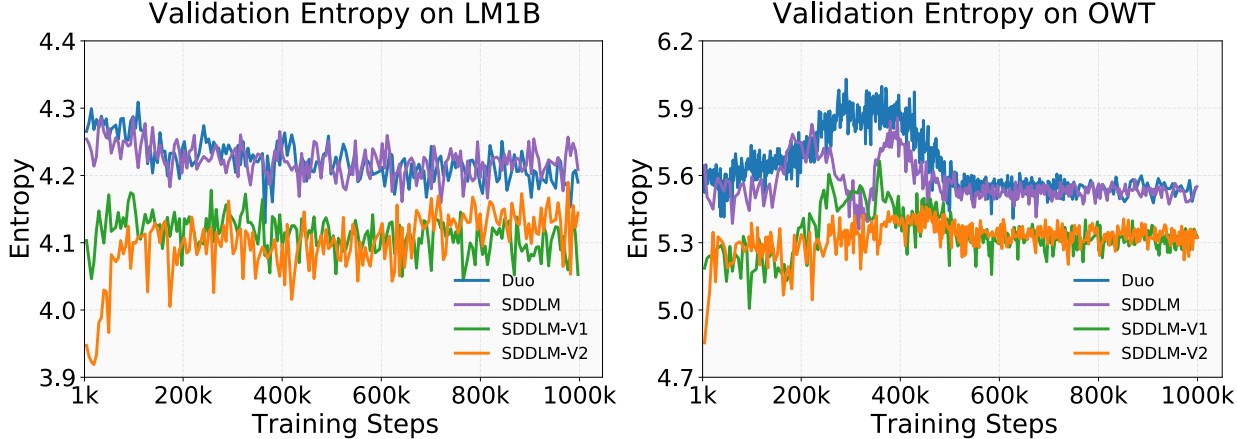

*Figure 5.* Validation entropy of different models over training steps.

|  | ARC-e | BoolQ | Hellaswag | Obqa | PIQA | RACE | SIQA | LAMBADA |
|---|---|---|---|---|---|---|---|---|
| Duo | 42.85 | **60.55** | 34.47 | 17.00 | 63.11 | 31.39 | 37.82 | 17.02 |
| SDDLM | 45.08 | 60.34 | 34.60 | **18.20** | **64.42** | 30.05 | 37.56 | 26.53 |
| SDDLM-V1 | **45.29** | 59.66 | **34.81** | 18.00 | 64.36 | **31.58** | **38.18** | **31.75** |

*Table 4.* **Evaluation of our larger (1.1B) models.** We scale up our model and evaluate it following the methodology described in Section C.4. We assess its performance on a diverse set of QA tasks as well as next-word prediction to comprehensively evaluate the capabilities of the larger models. Large values represent better performance in the table.

parable to the state-of-the-art method Duo, while outperforming other baseline approaches. Note that our model does not directly optimize the upper bound of the log-likelihood, whereas Duo explicitly optimizes this objective, which closely matches the evaluation protocol used in Table 3 (i.e., using an upper bound rather than the true log-likelihood). Therefore, it is expected that our model does not outperform methods that directly optimize the evaluation objective, and we observe a slight drop in the metric, consistent with similar observations reported in Deschenaux & Gulcehre (2024), when optimizing a loss that differs from the evaluation metric. Importantly, this minor decrease does not affect performance on conditional language generation (Details are in Section 5.4), which is more commonly used as the final evaluation criterion for larger models. This may be because diffusion language models are typically evaluated using an upper bound on the log-likelihood rather than the true log-likelihood; consequently, even when the upper bound exhibits a slight decrease, the models may still achieve comparable performance on the true likelihood.

### 5.4. Conditional Language Generation

In this section, we investigate the performance of our proposed method on conditional generation with larger models, a core language modeling task that has remained largely unexplored for USDMs. The corresponding results are reported in Table 4. We find that, despite its lower computational cost and favorable scaling properties, our method

achieves conditional generation performance comparable to approaches based on more complex ELBO-style objectives, such as Duo. Moreover, SDDLM-V1, equipped with an anti-uniform distribution sharpening regularization, attains performance on par with prior methods while demonstrating notably stronger results on next-word prediction tasks (e.g., Lambada). These empirical findings validate the effectiveness of the proposed regularization in mitigating the uniform distribution corruption issue. Overall, this simple yet effective regularization offers a promising direction for scaling USDMs to larger models with improved performance. More results about CFG are put into Appendix C.3.

## 6. Conclusion

We have shown how to simplify the training objective of Uniform State Diffusion Models (USDMs). Inspired by the denoising loss used in continuous diffusion models, we show that this minimal objective achieves performance comparable to prior, more complex formulations. Building on this observation, we introduce a negative-gradient mechanism motivated by a self-supervised learning perspective, which further improves the model. Notably, although the estimated perplexity (PPL) derived from the ELBO increases, the quality of the generated samples improves substantially, highlighting a mismatch between ELBO-based metrics and generation performance in USDMs. Finally, we demonstrate that the proposed approach scales effectively to larger

models, with consistent empirical trends, underscoring its potential for efficient large-scale training in the future.

## 7. Limitations

The main limitation of this work is that our current experiments are conducted on models with up to 1.1B parameters. Therefore, the effectiveness of our method has not yet been verified on larger-scale models or on real-world, long-context agentic tasks, such as coding, which could more clearly demonstrate the model's capabilities. Moreover, prior Masked Diffusion Language Models have demonstrated strong potential for text-to-image and multimodal generation. In contrast, our current method is limited to text generation. Therefore, an important direction for future work is to extend and evaluate our method in multimodal generation settings.

## Impact Statement

The primary goal of this work is to advance Machine Learning. The main societal consequences of this work are likely to stem from improved performance and algorithmic efficiency gains, as is the case with most fundamental research on machine learning.

## Acknowledge

This work was supported in part by grants from the National Science Foundation (2226025) and the National Center for Advancing Translational Sciences and the National Institutes of Health (UL1 TR002014) and by the Center for Artificial Intelligence Foundations and Scientific Applications and the Institute for Computational and Data Sciences at Pennsylvania State University.

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

## A. Omitted Details of Derivations

### A.1. Derivation of Equation (9)

Firstly, the regularization term in Equation (8) can be written in the following term:

$$\text{KL}\left(\mathcal{U}\|p_\theta\left(\cdot \mid x_t\right)\right) = -\mathbb{E}_{\hat{x} \sim \mathcal{U}(V)}\log p_\theta\left(\hat{x} \mid x_t\right) - \log|\mathcal{V}|. \tag{12}$$

Based on this Equation, we can the final objective for Equation (9) by ignoring $\log|\mathcal{V}|$ without gradient:

$$\mathcal{L}_{\text{SDDLM-V1}} = \mathbb{E}_{\mathbf{x}_0,t,\mathbf{x}_t \sim \mathcal{D},\mathcal{U},q_t}\sum_{l=1}^{L}(-\log p_\theta(\mathbf{x}_0^l \mid \mathbf{x}_t) + \mathbb{E}_{\hat{\mathbf{x}}^l \sim \text{U}(\mathcal{V})}\log p_\theta(\hat{\mathbf{x}}^l \mid \mathbf{x}_t))\mathbf{1}\left[\mathbf{x}_0^l \neq \mathbf{x}_t^l\right]. \tag{13}$$

### A.2. Relations between $\mathcal{L}_{\text{SDDLM}}$ and ELBO Loss

We follow the idea of Zekri et al. (2026) and first introduce a latent variable $s = (\mathbf{x}_t, t)$, where $q(s \mid \mathbf{x}) = q_t(\cdot \mid \mathbf{x}_0^l; \alpha_t)$, as defined in Equation (3). Under this formulation, we define a latent-variable model with the joint distribution $p^\theta\left(x_0, s\right) = p^\theta\left(x_0 \mid s\right)q(s)$ by parametrizing the denoising network. A standard ELBO with Jensen's inequality can be obtained as:

$$\log p_0^\theta\left(\mathbf{x}_0\right) \geq \mathbb{E}_{s \sim q(\cdot|\mathbf{x}_0)}\left[\log\frac{p^\theta\left(\mathbf{x}_0, s\right)}{q\left(s \mid \mathbf{x}_0\right)}\right]. \tag{14}$$

Then, by using the definition of $p\left(\mathbf{x}_0, s\right)$, we can obtain the following objective:

$$-\log p_0^\theta\left(\mathbf{x}_0\right) \leq \mathbb{E}_{s \sim q(\cdot|\mathbf{x}_0)}\left[\log p_0^\theta\left(\mathbf{x}_0 \mid s\right)\right] + C, \tag{15}$$

where $C$ is a constant, and the term in the equation corresponds to the denoising loss defined in Equation (7). More details can be found in Zekri et al. (2026).

## B. Details of Hyperparameter

In this section, we put details of hyperparameters of training for Duo, SDDLM, SDDLM-V1, SDDLM-V2 in Table 5. We follow the settings used in Duo for fair comparison. Moreover, we set $w = 1.1$ for training and evaluation separately.

*Table 5.* Training Hyperparameters for small models (170M).

| Hyperparameter | Duo | SDDLM | SDDLM-V1 | SDDLM-V2 |
|---|---|---|---|---|
| Optimizer | AdamW | AdamW | AdamW | AdamW |
| Learning Rate | $3 \times 10^{-4}$ | $3 \times 10^{-4}$ | $3 \times 10^{-4}$ | $3 \times 10^{-4}$ |
| LR Schedule | Linear Decay | Linear Decay | Linear Decay | Linear Decay |
| Warm-up Steps | 2500 | 2500 | 2500 | 2500 |
| Decay Rate $\beta_1$ | 0.9 | 0.9 | 0.9 | 0.9 |
| Decay Rate $\beta_2$ | 0.999 | 0.999 | 0.999 | 0.999 |
| Weight Decay | 0 | 0 | 0 | 0 |
| Global Batch Size | 512 | 512 | 512 | 512 |
| Training Steps | 1000k | 1000k | 1000k | 1000k |
| EMA Decay | 0.9999 | 0.9999 | 0.9999 | 0.9999 |

## C. Additional Details of Experiments

### C.1. Details of Datasets

In this section, we provide an overview of the benchmarks used in the Experiment section.

**WikiText** (Merity et al., 2016). This dataset consists of high-quality Wikipedia articles and is commonly used to assess general language modeling performance.

*Table 6.* Training Hyperparameters for Large models (1.1B).

| Hyperparameter | Duo | SDDLM | SDDLM-V1 |
|---|---|---|---|
| Optimizer | AdamW | AdamW | AdamW |
| Learning Rate | $2 \times 10^{-4}$ | $2 \times 10^{-4}$ | $2 \times 10^{-4}$ |
| LR Schedule | Linear Decay | Linear Decay | Linear Decay |
| Decay Rate $\beta_1$ | 0.9 | 0.9 | 0.9 |
| Decay Rate $\beta_2$ | 0.95 | 0.95 | 0.95 |
| Weight Decay | 0.1 | 0.1 | 0.1 |
| Global Batch Size | 256 | 256 | 256 |

**Lambada**. (Paperno et al., 2016) This dataset focuses on long-range dependency modeling, requiring the prediction of the final word in a passage given a broad context.

**AG News**. (Zhang et al., 2015) This dataset is a news corpus covering multiple topic categories and evaluates models' ability to model news-style text.

**ArXiv and PubMed**. (Cohan et al., 2018) These datasets contain scientific articles from their respective domains and are used to benchmark language modeling performance on long, technical documents.

For AG News and Scientific Papers (PubMed and ArXiv), we apply both the WikiText and One Billion Words detokenizers. Since the zero-shot datasets follow different conventions for sequence segmentation, we wrap all sequences to a length of 1024 tokens and do not insert end-of-sequence (EOS) tokens between consecutive segments.

**ARC-Easy.** (Clark et al., 2018) A subset of the AI2 Reasoning Challenge that concentrates on elementary-level science questions, designed to evaluate a model's reasoning ability based on fundamental scientific concepts.

**BoolQ.** (Clark et al., 2019) A yes-or-no question-answering dataset designed to evaluate a model's ability to answer questions based on a given passage.

**HellaSwag.** (Zellers et al., 2019) A dataset assesses the model's commonsense reasoning ability by completing a given sentence with one of four options.

**OpenBookQA.** (Mihaylov et al., 2018) A question-answering dataset modeled after open-book exams, designed to assess a model's understanding of a subject by requiring multi-step reasoning and the integration of additional commonsense knowledge.

**PIQA.** (Bisk et al., 2020) Physical Interaction Question Answering is a dataset that evaluates physical reasoning ability by asking models to select the best solution to problems involving everyday physical scenarios..

**SIQA.** (Sap et al., 2019) Social Interaction Question Answering is a commonsense reasoning benchmark that presents scenarios requiring models to reason about social interactions and the motivations underlying human behavior.

**RACE.** (Lai et al., 2017) ReAding Comprehension Dataset From Examinations was designed to evaluate reading comprehension ability through understanding and interpreting high school–level texts.

**LAMBADA.** (Paperno et al., 2016) A dataset designed to evaluate models' text understanding capabilities through a final single-word prediction task based on a given context.

### C.2. Details of Baselines

In this section, we provide detailed descriptions of baselines in experiments.

**Duo**. (Sahoo et al., 2025) This method improves USDMs by leveraging their connection to Gaussian diffusion, introducing Gaussian-guided curriculum learning for efficient training and discrete consistency distillation for fast few-step generation.

**SEDD**. (Lou et al., 2023) It introduces score entropy to extend score matching to discrete domains, enabling effective diffusion-based language modeling with strong perplexity and generation quality compared to both prior diffusion and autoregressive models.

|  | BoolQ | Hellaswag | Obqa | PIQA | RACE | SIQA | LAMBADA |
|---|---|---|---|---|---|---|---|
| Duo-CFG | **62.20** | 36.00 | 18.80 | 63.22 | **31.67** | 37.10 | 20.53 |
| SDDLM-CFG | 62.11 | 36.09 | **20.00** | **64.80** | 30.72 | 37.87 | 32.60 |
| SDDLM-V1-CFG | 62.17 | **36.22** | 19.00 | 64.47 | 29.86 | **38.95** | **34.70** |

*Table 7.* Additional Results for CFG methods on 1.1B models.

---

**Algorithm 1** SDDLM & SDDLM-V1
___
**Input:** Epochs $N$, trainable parameters $\theta$
**for** $i = 1$ **to** $N$ **do**
   $\mathbf{x}_0 \sim q(\mathbf{x}_0)$
   $t \sim \text{Uniform}[0, 1]$
   $\mathbf{x}_t \sim q_t\left(. \mid \mathbf{x}_0^l; \alpha_t\right), \hat{\mathbf{x}}^l \sim \mathcal{U}$ (for SDDLM-V1)
   Take gradient descent step on
   $\nabla_\theta \mathcal{L}_{\text{SDDLM}}$ or $\nabla_\theta \mathcal{L}_{\text{SDDLM-V1}}$
**end for**
**return** $\theta$
___

**UDLM**. (Schiff et al., 2024) This work extends classifier-based and classifier-free guidance to discrete diffusion models and introduces uniform-noise diffusion with a continuous-time training objective, enabling strong controllable generation.

**PLAID**. (Gulrajani & Hashimoto, 2023) This work improves the likelihood performance of diffusion language models via algorithmic advances and scaling law analysis.

### C.3. Additional Experiments

In this section, we present additional results for CFG applied to different algorithms, as shown in Table 7. We observe that CFG consistently improves performance across these methods. In particular, our simple loss formulation achieves comparable performance on QA tasks and superior results on next-word prediction. Moreover, incorporating our proposed negative-gradient regularization further enhances performance on next-word prediction tasks.

### C.4. Details of Training and Evaluation

**Training**. Both algorithms are summarized in Algorithm 1. They are deliberately simple: training proceeds by sampling either positive examples only, or both positive and negative examples (for SDDLM-V1), and optimizing the model under the corresponding objectives. Moreover, we adopt curriculum learning (Bengio et al., 2009), in which the model is trained on increasingly complex data, starting from simpler, easier-to-denoise noise patterns and gradually progressing to more challenging ones. Following Sahoo et al. (2025), this strategy has been shown to be effective for USDMs, and we adopt this curriculum to accelerate training. Notably, when scaling to larger models with 1.1B parameters in our experiments, our method can be readily applied and demonstrates strong potential for efficient scaling.

**Sampling and Evaluation**. For unconditional sampling, we iteratively sample from Equation (5) using the trained model, starting from a uniform distribution that randomly samples token sequences from the vocabulary. For conditional generation, we compute the model likelihood as defined in Equation 4 and apply it to QA tasks.

## D. Generation Examples

To ensure correct LaTeX rendering, we manually process the generated text following Duo (Sahoo et al., 2025):

1. Curly double quotes (`\u201c, \u201d`) replaced with "

2. Em dashes/en dashes (`\u2014, \u2013`) replaced with – or -

3. Soft hyphens (`\u00ad`) removed (or replaced by a normal hyphen where it makes sense)

4. Any other special characters replaced with a suitable ASCII approximation

Samples from model on OWT obtained from SDDLM. Preplexity=77.07, Entropy=5.53.

```
<|endoftext|>That does not mean, though, that you, like most people, can buy
↪ into a gigantic skill set. Nothing can inherently distinguish one NBA team
↪ from another in its caliber level or from any of those other building blocks
↪ involved. I know how I feel that way. I know countless powerhouses that didn't
↪  get the right players before the deadline, and compete for talent cutting
↪ talent without having them around. I see the aging stud handed new shoes,
↪ followed by an attempt to obtain a package of even greater caliber knowing
↪ their injuries might be over. But the blockbuster also gets its own haul.

That's just a rough copy of my estimation: the $212.05 million salary the
↪ Celtics announced Monday night hardly spooked this fight. It wouldn't
↪ necessarily change the league differently. I can tell now from all four years.
↪  But then there are the Titans. Something on the list would seem pretty tall:
↪ James 51, 14, Curry 20, 27, 17. Wade, his first year in the NBA at some level.
↪  That would 2,675800. This is not really long for long odds; his career salary
↪  was $100,000. He has been waiting in years. At the start of his career, he is
↪  shot 12-16 out of the field from a .940 rating. During the league's league
↪ semifinals playoff series Sunday against Brooklyn, he has recorded streaks of
↪ 15 points, 14 rebounds and 9 assists in 52 minutes.

Check out The Burn Tens: Snowpierizing Boston Startled Radio 4 World Center Live.
↪  Wade Has Missed the NBA Finals, Not MadeOne. I'm the judge: how does making
↪ room for an 11-12 season that could translate into outpacing more elite teams?
↪  For the Celtics, we are playing someone that looks to be part of a Western
↪ Conference champion in his final year before a trip to Italy, just before
↪ taking another shot in a much bigger city with a bright future. Durant, I take
↪  a time to note, had numerous Finals appearances complete following Westbrook'
↪ s arrival (I often did, at least until his release from LA), but was less
↪ popular than other teams. Still though, when you combine the sheer amount of
↪ playing time over the last three seasons, I can see why it would have taken
↪ him months to accomplish. I realize a bit of the age in decline, having
↪ happened with James, that this solo candidate is not intent on pulling off the
↪  championship trick. We get to that. If he isn't back when the doors get open
↪ - which he may do, with Minnesota possibly still stepping foot in the
↪ postseason - teams potentially need to take some notice. (I think they should
↪ make moves.) When I'm ready to do so, I wouldn't advocate the second Wade
↪ development unless there amounted to a buildup. They agents presumably playing
↪  games during that same time frame, or likewise at other stages - I don't even
↪  have time yet. I expect him to get reps on the big market from other teams,
↪ but I definitely start to point out the possibility of a franchise seeking
↪ more active franchises to sign him.

Which sort of scenario would it mean if the Celtics agree upon it?Previous Pick:
↪  Shane Battier (FCI Cavaliers). I think the Celtics should acquire Wade as a
↪ free agent July 4. Otherwise, I'd still see them offering as much as $44
↪ million deal ($48.5 million, $18.3 million guaranteed salary and $6.82 million
↪  guaranteed cap!). Here is my take - asreported by Jeremy Silver of pero.com
↪ for the five scenarios.<|endoftext|>
```

Samples from model on OWT obtained from SDDLM-V1. Preplexity=45.18, Entropy=5.31.

<|endoftext|>Baylor head football coach Bernie Byrne will be on the sideline
↪ for the Pac-12 college football game Saturday, which is set to start at noon.
↪ The most extensive game, Baylor is the second of Big 12 college football
↪ games in its first history. The Division II is enjoying this year's third
↪ consecutive losing streak. The Bears, breaking a 3-0 victories over the Pac
↪ -12's top-ranked USC and Oregon Sunday, lost Steve Spurrier to a 35-10 home
↪ loss to Missouri-&M in 2014. The head coach is retired, and his season has
↪ been overshadowed by Baylor's best second-ever NCAA campaign. The coach, who
↪ re-signed on June 23, is back for his 13th as a pro and is the No. 1 active
↪ coach of all time. "I was so excited to play the part here," said Byrne, whose
↪ daughter Cl.C. Tyler will play Saturday. "What this game brings us is to
↪ witness not only the national men's game, but we're now totally at Baylor,
↪ crazy. Our teams have looked good, there are four games of winning which is
↪ unprecedented."

    The family loves the opportunity to host the games and last year's win. We'
↪ re lucky to be before Saturday, Baylor's home opener." The schedule has Baylor
↪ saddled ahead of the Big 12 schedule. It includes preseason games from the
↪ Pac-12 plays Oregon and Notre Dame, and BCS tournament games for the entire
↪ 2015-16-20 Big Ten season. The final three-seed conference qualifies for the
↪ National Championship and covers the Pac-12. Normally this season, Texas Tech
↪ and Penn State would only require Stanford to host two of the top five. Texas
↪ hosts Oklahoma, No. 8, Duke, Washington and A&M, both in the 2014-14. It's
↪ also a tough football team, physically experienced. Nebraska coach Don
↪ Lombardi was a major factor in the game, helping throw the quarterback and
↪ senior receiver Drew Bischak the 3-1 win over the Huskers.

    There'll be one of Saturday's games on Monday, followed by an official news
↪ conference on campus Monday night through Wednesday and on Friday and Sunday
↪ morning. Texas first played in the 1984-85 season at Pins Hall. They only
↪ played one season before, and that is when Baylor went on a 3-0 win over
↪ Oregon. In 1990, the teams played each other twice on the same week. Each of
↪ those games began the Pac-12 tournament, for the first time in a decade
↪ without school top seeds. A.C. County will also be played. "We can always fill
↪ in the hole if we watch Baylor," Lombardi said in October about what this
↪ game offers to a school he's watched 43 seasons. "In the Big 12, every week,
↪ we expect all games to be played in the top schools. We're excited and shocked
↪ about it, but this will be our top 10 player's many moments."

    So much of its talent and overall strength in the Pac-12 is a surprise. The
↪ Huskers have been problematic, before returning to leave the season undefeated
↪ . In the Pac-12 West Conference Championship, despite the perception of the
↪ program over them as "safe" in the Big 12, the Big Ten and Pac-7 will host
↪ their first game, and will have two conference series games since Oct. 13 for
↪ the first time since 2002. Indiana beat Nebraska at home in its inaugural
↪ season. "We'll take to shutting them down 1-2," Anthony Soezzo said Friday. "
↪ If they were to do that, then I'm sure it would be upset for the sport and the
↪ country and Minnesota to play a national title against them."\n\nFor the
↪ Maryland program this year is a 'State-only' appearance in the Big 12 and in
↪ school history. The Terps will be faced Tuesday by the Air Force. An
↪ unblemished offense heading play Deon Jaxreli returns from injury and goes
↪ down with former three-star quarterback Cody Kessler. The key national
↪ discussion will be the Heisman Trophy race, presented by the ESPN College
↪ Football Playoff. The Utes last year among Big 12 schools for wins and total
↪ offense, and are also expected to host the final 3 SEC game of the regular
↪ season in August. The Tar Heels will play at Iowa State on Friday. Other
↪ regular games could take place on Saturday, Sept. 27, at 1 p.m on ESPN. The
↪ host is announced. The Irish have a 76-11 conference record (11 W, 2013) and a
↪ 31-23 conference home record over FCS football opponents (43-8, 2013), and
↪ will win on Tuesday. The Ols<|endoftext|>

Samples from model on OWT obtained from SDDLM-V2. Preplexity=50.05, Entropy=5.33.

```
    <|endoftext|> I just can't be bothered by it." Aside from Kelt's work,
↪ others don't want to run the Marlins. But other sports families have been able
↪  to be so bold. Soldiers, he says, and football players. Fitness classes and
↪ baseballs. All of it's to themselves that should be taken from. It doesn't
↪ work. "The most important part of it is, just how it's so easy to sell them a
↪ 'life by choice' and I don't want to have to write about it, I want to ask,
↪ why do you want to live with that price thing, why?" he said. "I don't think
↪ that's a gold standard for a job. It's not."

    "No matter who we have and money we get, just this work and this supportive
↪ community," he says. "Yes, the way I'm dealing with that, its 'saying.' That's
↪  my work and makes me money. That's my life." Last Thursday night, he talked
↪ about things in his life that didn't to have a career. The things he didn't
↪ know possible. The things inside him as his relationship with his mom or
↪ drawing manga or working on his second spell against him because of his job at
↪  his new job or a chance reading that he spends most of the time, plays tennis
↪ , can pay for, eat insurance and not get a cigar. "I decided that I want the
↪ life that I have, and the things that I am able to do these is how bad I'm and
↪  that it will be the rest of my life."

    I don't think how it's going to last, and what doesn't) what isn't the same
↪ ," he told the crowd. " 'Although sometimes what I do is wrong but every time
↪ it's the same is ' Like every time I go straight to lunch.'

    "He's been diagnosed with bipolar disorder and depression, but suffered from
↪  several accidental overdose, violence 35 years ago and trouble shooting in
↪ his career. He may not be focused on it, but he's trying, and his chances can
↪ be exacerbated by caution about using medications or a stimulant.

    "I can't be sure how the patient came over to take me, if I did them," Kelt
↪ said, "but I can definitely confirm it whatever the situation with the
↪ psychiatrist, whatever it might be." That's enough. There aren't many other
↪ things that he doesn't want to try in his life. But he has a life in his life,
↪  just by the way he has money and he has built a strong sense of survival and
↪ set out to come from there. There won't be any more crisis, he says. But he
↪ has had little contact with longtime free agent Mike Dargent, who joined his
↪ Cubs team a few years ago, after he took a job elsewhere, simultaneously
↪ contributing "a new home life" and learning how to switch between baseball and
↪  civilian sports.

    "When my new agent told me that I could get a good job and I made baseball
↪ my sport, it was the best thing I had known about things that I could do. Then
↪  Zach came in and I said, 'That is it. What can we do from here?' He wanted me
↪  to do it, which I wound up doing every day with Steve DeSantos. He's full
↪ grown in our people. Aron Arias, too." They have a 15-year-old son, Addison,
↪ who spent four big years with teammates Jim Lylesak and Eddie O'Neal, who have
↪  been playing shortstop since he was young. "It's my hand in the wind trying
↪ to be one of the stretch-threes I live with," said Dargent, when he was a 4-in
↪ -4-inch at his local high school and crossing bars. Locomack, whose has also
↪ been tracking his progress, though he's grown used to studying the art of
↪ baseball and craps and the point of exchange has to extend beyond baseball.

    "I messed up," he once said. "I tried to throw a hole it up and I made a
↪ mistake and He missed. I thought, you can't do what you get hurt with that
↪ thing, you think you'll do something a bit better. When he missed, you don<|
↪ endoftext|>
```

Samples from model on OWT obtained from Duo. Preplexity=80.43, Entropy=5.55.

```
    <|endoftext|> just discover a new position. Something reaches consensus. Go
↪ part Which football clubs did you admire during life at higher level? Well,
↪ five of them. Sometimes I decided to strengthen club, but also quickly. It was
↪  the really club in which I really started my career in my own league. I had
↪ never really played with Fernando Torres at Real Madrid when they arrived,
↪ just nice couple and a great time. You became the first Portuguese to qualify
↪ for the Premier League. Now did the only thing to stop you start breaking
↪ between the lines?

    Yes. You wouldn't think in the last two minutes it was a rational play. I
↪ was in a panic which was why he was throwing an almost-ball. I caught a pass I
↪  scored. I'm just whipping the ball out high on the to the pitch. Then the
↪ other one was making a punt in midfield and I took off.\" But at almost
↪ midnight. I'm very focused with my dreams and it was going to matter a lot in
↪ the rest of my life when I get involved again, but not until the end of the
↪ day. How would you conclude what happened to Andreco? Well, I think setting
↪ the example was beating him. Dark opportunities would be extremely difficult
↪ moments. I also think it wasn't an easy thing. That's why I had a season. I
↪ had to maintain myself in the front five because I was not always the best. Of
↪  course, of course that he would cause problems were impossible if not careful
↪  but I always wanted to win that position, right – until he punished me there.
↪  Tteno tap, and a lot of things happens while you are playing in that position
↪ . I tried to learn how to decide the ball in one direction quickly, rather
↪ than being down forward and trying to go attacking parts of the pitch. If you'
↪ d made a mistake operating inside of the centre half it was a nail in the
↪ coffin. Now I won't escape, then, to ask another question: what have you to
↪ gain from joining Bara? I feel really mature and succeeding very much. I was
↪ only 17 years old and my first thing priority was to get my demotion back.
↪ Everyone else pushed me into the team and then took us away. Absolutely but
↪ terms like being one of many players with transfers from the Europeans to
↪ Milan to Porto and Barcelona to Iniesta? No, actually I wanted to join Bara
↪ and it was told by Matala, at centre-back, understand that and understand what
↪  Barcelona wanted. Barcelona, picked me quickly and immediately mixed me as an
↪  important player to us.

    It got Sofia Bogarde a lot of queries: \"Has Drogba Barcelona before?\" \"
↪ Barcelona would rather never have invited you onto the pitch.\" \"I don't know
↪  if any of the rumours are true.\" But don't forget playing Catalunya! For me,
↪  I have got a perfect year in my football adventure. I am glorious: I'm not
↪ here to win the league or win trophies, but it's everything for Spain, and
↪ there is one thing to remember about here, what this country is used to.
↪ Cologne was lucky. It was three drops of rain, the sun and a change in air.
↪ The weather was like and was very kind and pleasant. The first ball went out
↪ of the tunnel. We were rewarded at minute of 7 when he came back, as the first
↪  team. We brought all the Barcelona lads, but we didn't accept that we had to
↪ play under Mourinho. Partnering was a lot of things that got you big but what
↪ are you thinking in those days? Now Bar\u00e7a went forward and clearly could
↪ beat us at any time with a Monaco side. At times we made a lot of mistakes in
↪ the game. Our manager gave us a style that we provide well. We have been
↪ enjoying success in style that Chelsea have developed. It's not because we
↪ cannot win in the Premier League. So it is immoral to let Barcelona win with
↪ that style. It was from our problems and how they covered the style, retracing
↪  it, that we realized that we had won. How did you think of their tactics I
↪ just always wanted to have a better relationship with them but we had a good
↪ relationship with the Portuguese!\" \"Do you know they are creeping me out on
↪ the fly from the pitch with mismanagement?\" \"We had to work with them.\"
↪ They won the Premier League over us, in the divisions and respected us. Plain
↪ and simple. We played in the SLC for 18 seasons. From 1970 to 1983 was a very
↪ cohesive team and we really believed as a team that we can defend the game
↪ well. It is a formation, that is, a team with midfield, big backs and
↪ everybody sharing the ball. It's decent in the box and turning around, and yet
↪  on the other side, we do fight<|endoftext|>
```

