# OpenReview forum: "Simple Denoising Diffusion Language Models"
_ICML.cc/2026/Conference — ICML 2026 regular_

### Official Review · Reviewer_c2pw · 2026-03-09

**Soundness:** 2
**Presentation:** 3
**Significance:** 2
**Originality:** 3
**Overall Recommendation:** 4
**Confidence:** 3

**Summary:**

The manuscript studies how to simplify the training objective of Uniform State Diffusion Models (USDMs) for language generation. The manuscript focus on the central question of whether a simpler denoising-based loss can replace the complex objective typically used in USDMs while maintaining strong generation performance.

**Compliance With Llm Reviewing Policy:**

Affirmed.

**Key Questions For Authors:**

* The paper shows that SDDLM-V1 improves generation quality while sometimes worsening ELBO-based PPL (e.g., Figure 4). Could the authors provide a deeper explanation of this mismatch between perplexity and generation performance, and clarify under what conditions this phenomenon occurs?
* The experimental results in Table 4 show that SDDLM variants underperform Duo on BoolQ, and that SDDLM-V1 performs worse than SDDLM on OBQA and PIQA. Could the authors analyze the potential reasons behind these behaviors and discuss whether they are related to task characteristics or model design?
* The paper emphasizes improved efficiency during training, but explicit inference-time comparisons are missing. Could the authors report generation latency or tokens-per-second comparisons against prior diffusion language models (e.g., Duo) under the same sampling steps?

**Limitations:**

Yes

**Strengths And Weaknesses:**

# Strengths
* The proposed objective is conceptually simple and well-motivated by the denoising paradigm used in continuous diffusion models.
* The paper provides empirical evaluations across multiple datasets and model scales (170M and 1.1B parameters).
* Ablations between SDDLM and SDDLM-V1 help illustrate the effect of negative gradients and regularization.

---
# Weakness
* The authors should provide a deeper analysis of the results in Figure 4, particularly explaining the relationship between perplexity (PPL) and generation performance for SDDLM and SDDLM-V1. It would be helpful to clarify why SDDLM-V1 shows different PPL behavior while achieving improved generation quality.
* The authors should include a more detailed discussion of the results in Table 4. In particular, it would be useful to analyze why the SDDLM variants underperform compared to Duo on BoolQ, and why SDDLM-V1 performs worse than SDDLM on tasks such as OBQA and PIQA. Such analysis could help better understand the strengths and limitations of the proposed approach across different task types.
* The paper does not report explicit inference latency or generation time comparisons with other models (e.g., Duo or other diffusion-based language models). Providing concrete wall-clock inference time or tokens-per-second metrics under different sampling steps would better demonstrate the practical efficiency of the proposed method.

---

> ### Author Rebuttal · Authors · 2026-03-30
>
> We gratefully appreciate your time in reviewing our paper. We would like to clarify some misunderstandings regarding our approach.
>
> **Q1. The authors should provide a deeper analysis of the results in Figure 4, particularly explaining the relationship between perplexity (PPL) and generation performance for SDDLM and SDDLM-V1.**
>
> **A1.** Thanks for suggestions!  We first clarify metrics to address some misunderstandings. Gen PPL shows generation quality, evaluated using GPT-2 large. PPL reported for USDMs is computed as $\exp(\mathcal{L}^l_{\text{USDM}}/L)$, following prior work, since exact likelihood is intractable for diffusion models and ELBO approximations are used instead. Accuracy is reported in Table 4 for real-world QA tasks.
>
> This suggests that PPL is not strongly correlated with generation quality, as it is based on an ELBO approximation (Duo's loss), which only provides an upper bound on the true PPL rather than the exact value. Therefore, it is expected that Duo achieves better PPL, as it directly optimizes this objective, while our method does not; nonetheless, our method still shows consistent improvement trends. In contrast, Gen PPL is more closely aligned with generation quality, as it reflects the behavior of large models such as GPT-2, and prior work [1] also notes that PPL may not correlate well with generative performance. **More importantly, the accuracy reported in Table 4 evaluates real-world QA tasks, and performance on this metric is known to be concrete and generalize well to larger models (e.g., 7B), as shown in prior MDM work.**
>
> **Q2. The authors should include a more detailed discussion of the results in Table 4. In particular, it would be useful to analyze why the SDDLM variants underperform compared to Duo on BoolQ, and why SDDLM-V1 performs worse than SDDLM on tasks such as OBQA and PIQA.**
>
> **A2.** Thanks for your suggestions! We find that SDDLM-V1 and SDDLM-V2 achieve comparable results on QA datasets such as OBQA and PIQA, with only small differences (0.2 and 0.006), making it hard to conclude that SDDLM-V2 performs worse. A similar trend is observed when comparing SDDLM with Duo (0.21 on BoolQ), making it difficult to conclude that SDDLM performs worse than Duo on these datasets.
>
> More importantly, on LAMBADA—focused on next-word prediction and closely tied to real-world generation—SDDLM-V2 significantly outperforms SDDLM-V1, suggesting that the sharper distributions induced by our negative gradient yield more confident predictions and improved generation performance. In terms of overall performance, the average scores are 34.65 for Duo, 39.59 for SDDLM-V1, and 40.45 for SDDLM-V2, demonstrating that our method outperforms Duo.
> **Finally, we would like to respectfully emphasize that the reviewer should not overlook the efficiency of our algorithm when evaluating overall performance, as our loss function is both simple and computationally efficient, as demonstrated in Figure 2.** We believe this efficiency, along with our contribution to simplifying USDM training and unifying it with MDM objectives, is an important aspect that should not be overlooked.
>
>
> **Q3. The paper does not report explicit inference latency or generation time comparisons with other models (e.g., Duo or other diffusion-based language models). Providing concrete wall-clock inference time or tokens-per-second metrics under different sampling steps**
>
> **A3.** Thanks for your suggestions! We report results for different sampling steps alongside their corresponding wall-clock inference times in the table below:
>
> | Steps                         | 8 | 16 | 32  | 64| 128  | 256| 512 | 1024 |
> |--------------------------------|----------|----------|----------|----------|----------|----------|----------|----------|
> |       MDMs   |   0.39s  |   0.88s  | 1.76s     | 3.61s | 7.65s     | 14.77s     | 29.54s     | 58.96s |
> |  Duo         |  0.41s   |    0.93s  |   1.80s  | 3.77s  | 7.45s     | 14.85s     | 29.72s     | 58.99s |
> |  SDDLM         |    0.42s  |    1.02s  |    1.83s  | 3.65s  | 7.47s     | 14.72s     | 29.60s     | 58.91s |
>
> For both Duo and SDDLM, we use the sampling method in Equation (5), resulting in similar sampling speed and the same number of steps as in masked diffusion models (MDMs). Notably, sampling with 16 steps (0.93s) already outperforms Duo with 1024 steps (58.99s), demonstrating a significant improvement in efficiency.
>
> [1] Beyondautoregression: Fast llms via self-distillation through time

---

> > ### Author Rebuttal · Reviewer_c2pw · 2026-04-01
> >
> > Thank you for the effort put into the rebuttal. I will raise my score to Weak Accept.

---

> > > ### Author Response · Authors · 2026-04-01
> > >
> > > Thank you for your thoughtful response and for recognizing our work. We greatly appreciate your support.

---

### Official Review · Reviewer_jW1u · 2026-03-10

**Soundness:** 3
**Presentation:** 3
**Significance:** 3
**Originality:** 2
**Overall Recommendation:** 5
**Confidence:** 3

**Summary:**

The Unified State Diffusion Model (USDM) is expected to achieve rapid text generation due to its self-correction capability. However, its complex loss design and additional computations limit its scalability. To address this, this paper proposes a simplified denoising loss that only optimizes the marked tokens replaced by noise, achieving comparable performance to complex methods while stabilizing the training process. Additionally, a lightweight regularization term is introduced to alleviate output deviations, further enhancing the effect. Through pre-training on text datasets, the effectiveness and efficiency of this method are verified, and it is demonstrated to be applicable on larger-scale models as well.

**Compliance With Llm Reviewing Policy:**

Affirmed.

**Final Justification:**

I suggest accepting this article.

**Key Questions For Authors:**

(1) In equation (8), you introduced the regularization term with the aim of reducing the deviation towards a uniform output distribution and interpreted it from the perspective of contrastive learning as providing an exclusion gradient. However, from an information theory perspective, minimizing the KL divergence between the predicted distribution and the uniform distribution is equivalent to maximizing the entropy of the predicted distribution. This seems to present a seeming contradiction with your claim of encouraging the model to generate sharper and more discriminative label distributions. Could you clarify the exact mechanism behind this regularization term?
(2) Under the same model size and training data, how does your method compare in performance to the current mainstream state-of-the-art autoregressive models (such as the equally sized LLaMA-2 or OPT)?
(3) Could you conduct a more in-depth analysis of the pros and cons of the two negative sampling strategies: random sampling from the uniform distribution (V1) and using the noise version itself as negative samples (V2)?

**Limitations:**

The author only provided a brief "Impact Statement" at the end of the paper, generally mentioning that the work might bring about improvements in performance and efficiency, but did not delve into any potential negative social impacts, nor did he set up a dedicated chapter to systematically expound the limitations of the work. It is suggested that in the conclusion section or by adding a new subsection titled "Limitations and Future Work", the technical limitations of the method should be clearly listed, such as: the theoretical motivation of the regularization term can still be further explored, the numerical stability during training depends on the hyperparameter ε, and the sub-optimality of the negative sampling strategy (the performance differences between V1 and V2 on different datasets lack an explanation).

**Strengths And Weaknesses:**

Soundness: The theoretical and experimental analyses in the paper demonstrate the technical correctness and rigor of the method. It can be seen from the text that the author has conducted experiments through multiple zero-shot datasets and evaluation benchmarks, covering a variety of task scenarios. This extensive experimental design indicates that the author's evaluation of the method is rigorous and comprehensive. The proposed improvement method takes a simple denoising mechanism as the core and introduces innovative adjustments to the existing diffusion methods.

Presentation: The overall writing of the article is clear, and the structure is reasonable, enabling it to convey the methods and results in a complete and concise manner. Especially in the experimental section, the paper provides detailed descriptions of the data sets, explanations of the methods, and comparisons of the results, introducing the performance of different tasks and models. However, some terms or details (such as the specific mathematical definition and derivation of the negative gradient loss) may require further explanation to enable readers to more easily understand the core logic of the new method.

Significance: The issue of how to improve the performance of diffusion models in generation tasks through simple denoising mechanisms is one of the important research directions in the field of machine learning. Especially in the optimization of generative models, the author's work helps to enhance the quality and diversity of the generated content. The main contribution of the paper seems to focus on the progressive improvement of the technology rather than proposing a new framework, so its scope of influence may be relatively limited. The results of the paper are mainly in the academic community, and whether they can be well adopted or extended to practical applications still needs further verification.

Originality: The core denoising method of the paper is based on existing theories and has been improved by simple adjustments and the introduction of negative gradients to the diffusion model. This innovation is more reflected in the creative combination of ideas rather than a completely new design from scratch. Nevertheless, this gradual improvement provides an effective and cost-effective method for the existing technology.

---

> ### Author Rebuttal · Authors · 2026-03-30
>
> Thanks for your thoughtful comments! Please see our responses below:
>
> **Q1. However, from an information theory perspective, minimizing the KL divergence between the predicted distribution and the uniform distribution is equivalent to maximizing the entropy of the predicted distribution. This seems to present a contradiction with your claim of encouraging the model to generate sharper and more discriminative label distributions.**
>
> **A1.** Thanks for your question! We believe there may be some misunderstanding of this loss. We minimize the negative KL divergence, which is equivalent to **maximizing the KL divergence  (not minimize KL divergence)**, leading to sharper and more discriminative label distributions. We will further emphasize this in the revised version.
>
> **Q2. Under the same model size and training data, how does your method compare in performance to the current mainstream state-of-the-art autoregressive models (such as the equally sized LLaMA-2 or OPT)?**
>
> **A2.** Thanks for your comments! We conduct experiments on LLaMA-2-based autoregressive models with 1B parameters, same optimization steps and data and obtain the following results:
>
> | Datasets                         | RACE | SIQA  | BoolQ | ARC-e |
> |--------------------------------|----------|----------|----------|----------|
> |  AR        | 29.60     | **38.28**     | **59.69**     | 45.19 |
> |  SDDLM-V1          | **31.58**     | 38.18     | 59.66     | **45.29** |
>
> We observe that, under the same number of optimization steps, our method is comparable autoregressive models with similar size based on LLaMA-2.
>
> **Q3. Could you conduct a more in-depth analysis of the pros and cons of the two negative sampling strategies: random sampling from the uniform distribution (V1) and using the noise version itself as negative samples (V2)?**
>
> **A3.** Thanks for your suggestions! The purpose of this experiment is to explore the impact of different negative sampling strategies on model performance. Our findings are primarily empirical, aiming to provide qualitative insights rather than definitive conclusions. When using the noisy version of the input as negative samples, the model is encouraged to modify tokens during inference, which aligns with the fundamental denoising capability expected of diffusion-based models. This prevents the model from degenerating into a trivial solution where it simply preserves the input tokens without performing meaningful denoising. However, such basic denoising abilities may not be sufficiently learned on smaller and simpler datasets, such as LM1B. As a result, SDDLM-V1 demonstrates improved performance over SDDLM-V2 in these settings. Specifically, SDDLM-V2 promotes a sharper output distribution, leading to more confident predictions. On larger and more complex datasets, such as OWT, the model is better able to learn effective denoising behaviors. In these cases, SDDLM-V2 further enhances performance by producing sharper and more confident distributions, benefiting from the richer training data. This in-depeth analysis is all from empirical results and we will further explore this in the future work.
>
> **Q4. t is suggested that in the conclusion section or by adding a new subsection titled "Limitations and Future Work", the technical limitations of the method should be clearly listed**
>
> **A4.** Thanks for your suggestions! Regarding theoretical justification, our method is closely connected to the ELBO formulation of MDMs, which provides a principled understanding of our objective (see [1]). We have elaborated on this connection in detail in A1 and A2 of our response to Reviewer f3UC. In addition, the introduction of $\epsilon$ is a standard numerical stabilization technique for log operations. Since negative samples typically have extremely low probabilities due to the strong fitting capacity of transformers, directly applying the log can lead to very large gradients. By adding a small $\epsilon$, we prevent numerical instability and avoid abnormal parameter updates during training. We also provide demonstrations for negative sampling in A3. We will discuss this limitation section in the revised version.
>
> Importantly, we would like to respectfully emphasize that these limitations can be adequately addressed based on our previous responses and do not diminish the significance of our main contributions. **Our main contribution lies in simplifying the training objective for USDMs and unifying it with the objective used in MDMs, without requiring separate and complex ELBO derivations for USDMs.** Extensive experiments demonstrate that our method consistently outperforms, or at least matches, the performance of existing approaches. Overall, as illustrated in Figure 2, our method achieves both efficient and effective training, and it can be naturally extended to larger-scale models, demonstrating promising scaling potential for future work.
>
> [1] HowMaskMatters: Towards Theoretical Understandings of Masked Autoencoders

---

> > ### Author Rebuttal · Reviewer_jW1u · 2026-04-01
> >
> > The author has already provided an explanation for my doubts and supplemented the experiment I proposed.

---

> > > ### Author Response · Authors · 2026-04-01
> > >
> > > Thanks for your response! We greatly appreciate your further recognition of our work and your consideration in raising the score.

---

### Official Review · Reviewer_f3UC · 2026-03-10

**Soundness:** 2
**Presentation:** 2
**Significance:** 2
**Originality:** 3
**Overall Recommendation:** 4
**Confidence:** 3

**Summary:**

The paper proposes a new algorithm for training uniform state discrete diffusion models. The first main contribution consists in noticing that as one computes the objective on language tokens, one needs to only consider the tokens which are perturbed. The second contribution consists in introducing an additional repulsion term to the objective which encourages the tokens distribution to be far away from the uniform distribution. The paper also has experimental evidence that the proposed methods might be better than existing approaches in certain scenarios.

**Compliance With Llm Reviewing Policy:**

Affirmed.

**Final Justification:**

The authors addressed my concerns and I increase my score to 4.

**Key Questions For Authors:**

Can you provide theoretical justification for your losses ? Especially for the one with repulsion and with epsilon?

Can you explain the discrepancy in your experimental results?

Can you substantiate the claim you make in Section 5.3, "we observe a substantial improvement in the quality of generated samples"?

**Limitations:**

yes

**Strengths And Weaknesses:**

Soundness:

While the contribution by eq.7 is interesting, it is unclear how the introduced objective is related to the ELBO/likelihood and therefore is not theoretically justified. The empirical evidence suggests that in some cases it helps, but the improvements over the baseline are not consistent. Therefore, it is whether the methodology is sound.

The objective function introduced in eq.8/eq.9, while interesting, feels adhoc. It is not related to ELBO/likelihood and the authors even themselves point out that it is the case. While I like the idea of adding the repulsion term, it will significantly strengthen the paper if this design choice is justified. Moreover, as a minor point, one might want to add a KL not from uniform but from a prior distribution, where prior distribution can differ from uniform.

The approach with repulsion seems to decrease generative perplexity but it also decreases the entropy. Moreover, the Figure 4 shows that the perplexity of SDDLM-V1 is consistently worse than Duo. This inconsistency suggests that something contrived might be going on. I would appreciate if the authors explain this discrepancy. Furthermore, in Table 2, we see that SDDLM-V1 is better than SDDLM-V2 in one case but worse in another. Figure 3 suggests that SDDLM-V2 is consistenty worse than SDDLM-V1 and it is unclear which dataset is being used. Finally, Table 3 suggests that the proposed SDDLM approach seems to be overall worse than Duo, while Table 4 shows that SDDLM is sometimes better than Duo. I.e., it is worse than Due on LAMBADA in Table 3 but better in Table 4. This inconsistency creates a large confusion on what is going on. Finally, Figure 5 suggests that SDDLM-V1/V2 lead to consistently lower entropy, which might be problematic for distributions with large diversity.

The authors point out that "we observe a substantial improvement in the quality of generated samples", but this claim is not substantiated by any quantitative metric. In fact, the fact that the entropy is consistently lower might actually suggest that the model leads to lower diversity samples.

The authors use "epsilon" in eq.9 and say that without using the "epsilon", the method does not work. First, I think it will be cleaner if this epsilon is explicitly written up in the equation since this is what actually being used and this is crucial for making the method work. Second, the authors do not explain which epsilon they use in practice, nor provide any ablation on it. Third, it becomes even less clear what a method with "epsilon" is supposed to do. At the moment, it feels more like a hack to make the idea work, and therefore some theoretical insight would significantly help understanding why this is needed.

Presentation:

The main idea is clear, but the presentations of figures could be improved. When Figure 1 is introduced on the first page and later referred in Section 4.1, "Duo" and other methods are not introduced. The authors should add more information in the caption of the figure to make it easier to follow. See above regarding "epsilon", I think it should be explicitly written in the equations.

Significance:

I think the paper could be quite impactful if the design decisions were better justified and the empirical evidence had consistent results. At the moment, it is hard to say whether this idea really works in practice and whether it should be used by practitioners.

Originality:

I think the insight that one should compute the loss only on perturbed tokens is novel and is consistent with the work on continuous diffusion models.

---

> ### Author Rebuttal · Authors · 2026-03-30
>
> We appreciate the feedback and will clarify some misunderstandings of our method.
>
> **Q1. It is unclear how Eq.7 is related to the ELBO**
>
> **A1.** Our loss is inspired by the denoising view of diffusion models [1] and isn't related to ELBO in Eq.4. However, this doesn't mean it is unrelated to ELBO-based loss. The ELBO loss in MDMs has a similar structure, as it predicts tokens only at masked positions (Eq.10 in [2]), whereas our loss predicts tokens at perturbed positions. From this perspective, USDMs can be interpreted through a masked modeling lens, where each token is either perturbed or unperturbed—similar to masked or unmasked states. **Under this view, the ELBO derived for MDMs can be applied to USDMs, where predictions are made on perturbed tokens in USDMs (our Eq.7) and on masked tokens in MDMs (Eq.10 in [2]), thereby justifying a clear connection between our Eq.7 and the ELBO used in MDMs.** . Thus, our method unifies MDM and USDM objectives using the MDM ELBO, giving a simpler view without complex derivations for USDMs.
>
> **Q2. eq.8/eq.9 feels adhoc.**
>
> **A2.** We clarify that our method isn't related to the ELBO for USDMs in Eq.4. It is motivated by viewing diffusion models as self-supervised learning [1], and we enhance the denoising objective with contrastive learning, discussed in Section 4.3. Also, USDMs can be a special case of MDMs (discussed in A1). Prior work [3] has theoretically analyzed the limitations of masked prediction loss (the first term in Eq.8) and introduced uniform regularization (the second term in Eq.8). This regularization is closely related to contrastive learning (e.g., Eq.10 in [3]) and helps improve feature diversity. Together, these motivate our method and provide theoretical support. We will further add A1 and A2 in our revised version.
>
> **Q3. Prior distribution for KL can differ from uniform.**
>
> **A3.** We also use other priors, such as treating input tokens as negative samples in SDDLM-V2, and will show more priors in the future.
>
> **Q4. Discrepancy in experimental results.**
>
> **A4.** We first clarify metrics to address some misunderstandings. Gen PPL shows generation quality, evaluated using GPT-2 large. PPL reported for USDMs is computed as $\exp(\mathcal{L}^l_{\text{USDM}}/L)$, following prior work, since exact likelihood is intractable for diffusion models and ELBO approximations are used instead. Accuracy is reported in Table 4 for real-world QA tasks. Here are answers:
> -  **SDDLM-V1 worse than Duo in Figure 4:** This is because PPL is computed from the Duo loss, which our method doesn't directly optimize, so it is naturally worse than Duo. However, ELBO-based PPL is only an approximation and doesn't reflect true likelihood. Still, our method reduces ELBO with lower training cost, capturing the optimization trend. Importantly, Gen PPL better reflects true generation quality, while ELBO-based PPL may be slightly lower since it is only an approximation [4]. Table 4 reports accuracy on real-world QA tasks, where our model outperforms Duo on average. This metric is more concrete and can transfer to larger models (e.g., 7B) from prior MDM findings.
> - **SDDLM-V1 isn't always better than V2 in Table 2. SDDLM-V2 is worse than V1 in Figure 3 and unclear for datasets.**: For SDDLM-V2, we explore other prior distributions by sampling negative examples. Empirically, we find that SDDLM-V1 tends to perform better on more complex datasets, while SDDLM-V2 may be more effective on simpler tasks. We don't claim that one is universally superior, but rather aim to explore different formulations. Figure 3 is on OWT.
> - **SDDLM is worse in Table 3 but Table 4 shows that SDDLM is better:** We have addressed Table 3’s issue in Point 1. Notably, our model gains better avg. performance than Duo in Table 4; minor drops on certain datasets are within a comparable range. The results of LLAMDA reported in Table 3 and Table 4 use various metrics, as clarified earlier.
>
> **Q5. Generation quality isn't verified by quantitative metric. Entropy is lower.**
>
> **A5.** Generation quality is reflected by the Gen PPL in Figure 3. A slight decrease in entropy is acceptable given the improvement in Gen PPL, as also observed in [4] and prior work on MDMs.
>
> **Q6. Epsilon should be written up. Show epsilon or provide ablation.  less clear about epsilon.**
>
> **A6.** We will include this in the equations and specify $\epsilon = 10^{-4}$ in Appendix B. The model can't be trained for clean generation of this ablation with very large PPL, so we don't report metrics. Also, $\epsilon$ is a standard numerical trick—adding a small value before the log to avoid gradient instability from very small inputs—and no need for theoretical analysis.
>
> [1]Deconstructing denoising diffusion models for self-supervised learning
>
> [2]Simple and Effective Masked Diffusion Language Models
>
> [3]HowMaskMatters: Towards Theoretical Understandings of Masked Autoencoders
>
> [4]Beyondautoregression: Fast llms via self-distillation through time

---

> > ### Author Rebuttal · Reviewer_f3UC · 2026-04-02
> >
> > Thank you for your rebuttal.
> >
> > > We also use other priors, such as treating input tokens as negative samples in SDDLM-V2, and will show more priors in the future.
> >
> > Can you demonstrate results with these?
> >
> > > Under this view, the ELBO derived for MDMs can be applied to USDMs, where predictions are made on perturbed tokens in USDMs (our Eq.7) and on masked tokens in MDMs (Eq.10 in [2]), thereby justifying a clear connection between our Eq.7 and the ELBO used in MDMs. . Thus, our method unifies MDM and USDM objectives using the MDM ELBO, giving a simpler view without complex derivations for USDMs.
> >
> > Even if it has a similar structure as ELBO, does it actually mean that your objective corresponds to ELBO ? If so, can you please add a derivation which supports this claim?
> >
> > > The results of LLAMDA reported in Table 3 and Table 4 use various metrics, as clarified earlier.
> >
> > Could you please clarify?
> >
> > > Together, these motivate our method and provide theoretical support. We will further add A1 and A2 in our revised version.
> >
> > Is there an objective (i.e., lower bound on likelihood or similar) such that your method can be derived?

---

> > > ### Author Response · Authors · 2026-04-02
> > >
> > > Thanks for your reply and further questions! We would like to do the following clarification:
> > >
> > > **Q1. Can you demonstrate results with different priors?**
> > >
> > > **A1.** Thanks for your questions! We consider two choices of prior distribution in our framework. In SDDLM-V1, the prior is defined as a uniform distribution over the vocabulary (i.e., randomly sampled tokens). In contrast, SDDLM-V2 adopts a data-dependent prior by sampling directly from the perturbed tokens. These design choices are detailed in the implementation description in Section 5.1. We further present comprehensive empirical comparisons between SDDLM-V1 and SDDLM-V2 in Figures 1–5 and Table 2.
> > >
> > > **Q2. Even if it has a similar structure as ELBO, does it actually mean that your objective corresponds to ELBO ? If so, can you please add a derivation which supports this claim?**
> > >
> > > **A2.** Thanks for following up with questions! We can derive the ELBO from an alternative perspective by treating $s = (x_t, t)$ as a latent variable, where $q(s \mid x_0) = \rho(t), q_t(x_t \mid x_0)$, instead of modeling the entire diffusion path. The latter modeling the entire diffusion path is typically interpreted as a hierarchical VAE and is widely used for deriving diffusion objectives in Section 2.1 of [1], where $\rho$ denotes the uniform distribution over $[0,1]$. Under this formulation, we define a latent-variable model with the joint distribution $p^\theta (x_0, s) = p^\theta(x_0 \mid s) q(s)$ by parametrizing the denoising network. A standard ELBO with Jensen’s inequality can be obtained as:
> > >
> > > $\log p_0^{\theta}\left(x_0\right) \geq \mathbb{E}_{s \sim q\left(\cdot \mid x_0\right)}\left[\log \frac{p^{\theta}\left(x_0, s\right)}{q\left(s \mid x_0\right)}\right]$.
> > >
> > > Then, by using the definition of $p(x_0, s)$, we can obtain the following objective:
> > >
> > > $\log p_0^{\theta}\left(x_0\right) \geq \mathbb{E}_{s \sim q\left(\cdot \mid x_0\right)}\left[-\log p_0^{\theta}\left(x_0 \mid s\right)\right] + C$,
> > >
> > > where $C$ is a constant not related to $\theta$. We can observe that this ELBO corresponds to a reconstruction loss ($-\log p_0^{\theta}\left(x_0 \mid s\right)$) that includes both predicting the perturbed tokens (i.e., $L_{SDDLM}$). and the unperturbed tokens. As discussed in Section 4.1, including the unperturbed tokens may lead to trivial solutions, since these positions require the model to simply copy the input. Moreover, it is challenging for neural networks to learn when to preserve tokens and when to denoise it under randomly sampled corruption. Therefore, based on our empirical observations (simply reconstruction loss will lead to very large Gen PPL in practical), we derive the SDDLM loss by restricting prediction to perturbed tokens only, resulting in an objective that shares a similar structure with the MDM formulation. In summary, this builds the relation between ELBO and our loss $L_{SDDLM}$ by ignoring predicting unperturbed tokens. We will add this discussion in our revised version.
> > >
> > > **Q3. Could you please explain metrics used in Table 3 and Table 4?**
> > >
> > > **A3.** Thanks for your comments! PPL in Table 3 reported for USDMs is computed as $\exp(\mathcal{L}^l_{\text{USDM}}/L)$ (Duo’s loss), following prior work, since exact likelihood is intractable for diffusion models and ELBO approximations are used instead. Accuracy is reported in Table 4 for real-world QA tasks. **Table 4 reports accuracy on real-world QA tasks, where our model outperforms Duo on average. This metric is more concrete and can transfer to larger models (e.g., 7B) from prior MDM findings.**
> > >
> > > **Q4. Is there an objective (i.e., lower bound on likelihood or similar) such that your method can be derived?**
> > >
> > > **A4.** Thanks for your question! We think this question is closely related to Q2, which we have already addressed in detail. Our loss is connected to an upper bound of the likelihood; please refer to A2 for a more comprehensive explanation.
> > >
> > > Finally, we would like to emphasize that our method **is both simple and effective**, as demonstrated in Figure 2. Moreover, **the real-world results in Table 4 provide strong evidence of its potential to scale to larger models.**
> > > Our approach offers a new perspective for training USDMs, **moving beyond reliance on complex objectives by unifying the losses of MDMs and USDMs.** This unification also opens the door to hybrid training strategies that combine the strengths of both paradigms in the future, where USDMs focus on token editing and MDMs focus on token prediction. We respectfully emphasize that these contributions are central to our work and should not be overlooked when evaluating its significance.
> > >
> > > [1] The principles of diffusion models
> > >
> > > **Update:**
> > >
> > > Thanks for your further questions! As the discussion period nears its end, we'd like to ask if there are any remaining questions or points that require further clarification. If our responses have addressed your concerns, we would be grateful if you would consider updating your score accordingly.

---

### Decision · Program_Chairs · 2026-04-30

**Decision:**

Accept (regular)

**Comment:**

This paper proposes a simplified denoising loss for Uniform-State Diffusion Models (USDMs) that optimizes only noise-replaced tokens, along with a regularization term to prevent uniform output collapse. The key contribution is unifying USDM and MDM training objectives without complex ELBO derivations. All three reviewers converged to positive scores after rebuttal (5, 4, 4).

The simplified objective is well-motivated, computationally efficient, and achieves competitive performance with the more complex Duo baseline. Scaling to 1.1B parameters with QA improvements (Table 4) demonstrates practical potential.

Reviewer f3UC raised the most substantive concern: the theoretical relationship between the proposed loss and ELBO. The authors provided a clear derivation and metric clarifications; f3UC found these adequate and raised their score. Reviewers jW1u and c2pw marked their concerns as fully resolved. The authors should incorporate the ELBO derivation, explicit epsilon notation, and a limitations section into the camera-ready.